# Molecular Imaging of Angiogenesis in Oncology: Current Preclinical and Clinical Status

**DOI:** 10.3390/ijms22115544

**Published:** 2021-05-24

**Authors:** Alexandru Florea, Felix M. Mottaghy, Matthias Bauwens

**Affiliations:** 1Department of Nuclear Medicine, University Hospital RWTH Aachen, 52074 Aachen, Germany; aflorea@ukaachen.de (A.F.); matthias.bauwens@mumc.nl (M.B.); 2Department of Radiology and Nuclear Medicine, Maastricht University Medical Center, 6229HX Maastricht, The Netherlands; 3School for Cardiovascular Diseases (CARIM), Maastricht University, 6229HX Maastricht, The Netherlands; 4School of Nutrition and Translational Research in Metabolism (NUTRIM), Maastricht University, 6229HX Maastricht, The Netherlands

**Keywords:** oncology, angiogenesis, PET, SPECT, VEGF, RGD, NGR, fibronectin

## Abstract

Angiogenesis is an active process, regulating new vessel growth, and is crucial for the survival and growth of tumours next to other complex factors in the tumour microenvironment. We present possible molecular imaging approaches for tumour vascularisation and vitality, focusing on radiopharmaceuticals (tracers). Molecular imaging in general has become an integrated part of cancer therapy, by bringing relevant insights on tumour angiogenic status. After a structured PubMed search, the resulting publication list was screened for oncology related publications in animals and humans, disregarding any cardiovascular findings. The tracers identified can be subdivided into direct targeting of angiogenesis (i.e., vascular endothelial growth factor, laminin, and fibronectin) and indirect targeting (i.e., glucose metabolism, hypoxia, and matrix metallo-proteases, PSMA). Presenting pre-clinical and clinical data of most tracers proposed in the literature, the indirect targeting agents are not 1:1 correlated with angiogenesis factors but do have a strong prognostic power in a clinical setting, while direct targeting agents show most potential and specificity for assessing tumour vascularisation and vitality. Within the direct agents, the combination of multiple targeting tracers into one agent (multimers) seems most promising. This review demonstrates the present clinical applicability of indirect agents, but also the need for more extensive research in the field of direct targeting of angiogenesis in oncology. Although there is currently no direct tracer that can be singled out, the RGD tracer family seems to show the highest potential therefore we expect one of them to enter the clinical routine.

## 1. Introduction

Angiogenesis is an active process, which regulates the growth of new blood vessels from a pre-existing vascular bed and is crucial for any regenerative mechanisms and for the survival and growth of tumours. Under physiological conditions a fine-tuned regulation of pro- and anti-angiogenic factors expands the surrounding vascular network in the target tissue. However, in the case of solid cancers, there is a pathological misbalance in the favour of the pro-angiogenetic factors, which results in abnormal, leaky, new capillary formation, required for its rapid growth and metastasis [1]. Beyond its effects on tumour expansion, these immature capillaries facilitate tumour metastasis by providing an efficient route for the tumour cells to exit the primary site and enter the blood stream [2]. Therefore, pro-angiogenetic factors, including their respective receptors, are attractive targets in the molecular imaging of tumour vascularisation and vitality assessment.

Concentration gradient via diffusion may achieve a steady nutrient supply and waste product elimination only in tissues smaller than 2 mm in diameter. Henceforth, any tumour larger than this requires vascularisation for its subsequent growth. This is an important aspect, which is frequently overlooked, as both chemotherapy and imaging tracers that directly target the process of angiogenesis have a limited effect (if any) on primary tumours and metastases smaller than 2 mm. Thus, the detection capabilities of imaging probes targeting angiogenesis are restricted both by the spatial resolution of the scanner and by the tumour volume. Nonetheless, as most solid tumours are prone to acquire resistance to anti-angiogenic therapies [3,4], there is an unmet need for non-invasive targeted imaging of tumour vascularisation and vitality.

Computed tomography (CT) and magnetic resonance imaging (MRI) are in principle not ideal for assessing tumour angiogenesis. These techniques mainly evaluate parameters such as changes in tumour volume, morphology, or even tumour blood flow/blood volume and vascular permeability, but they cannot quantify changes in the tumour vasculature. However, there are significant advances being made, especially with MRI, that allow imaging of angiogenesis-correlated parameters. This is beyond the scope of the current review, but interesting applications may be found in the following reviews [5,6,7].

Molecular imaging, either via positron-emitting isotopes for imaging with PET or via single photon emitting isotopes for imaging with SPECT, makes use of radioactively labelled tracers that bind to specific targets. The technique of molecular imaging is significantly more sensitive compared to CT and MRI, allowing for the visualisation and even quantification of sub-nanomolar concentrations of targets. Both PET and SPECT imaging techniques rely on small amounts of “tracers”, which bind a specific molecular target. Both modalities comes with a long list of tracer parameters that need to be met, including good target to background ratio, fast clearance (preferably renal), suitable biodistribution, no toxicity, and high specificity for the specific target. Therefore, the field of molecular imaging is ever developing with improvements in the tracer, either via incremental improvements within a given class of compounds; or via “new” tracers that are used for targets that previously could not be detected.

The possible molecular mechanisms to combat angiogenesis in oncology have been previously described in many papers [1,4,8,9,10]. The scope of this review is to present the possible imaging options of tumour vascularisation and vitality via non-invasive molecular imaging, including potential PET and SPECT tracers to be implemented in the clinic.

The key question is to identify a suitable target for the PET and SPECT tracers. In adult tissues, under physiological conditions, there is a balance between the pro-and anti-angiogenic factors [11]. In situ tumours are usually separated from the vascularised peri-tumoural tissues by a basal membrane and blood vessels are rarely present in these lesions [12]. However, in the ever increasing hypoxic conditions created when reaching 2 mm, tumour cells start secreting paracrine factors, thus recruiting surrounding stromal fibroblasts, which will differentiate into tumour associated fibroblasts (i.e., CAFs) [13]. The CAFs in turn lyse local extracellular matrix via matrix metallo-proteases (MMPs), secrete inflammatory factors (e.g., interleukins) and start laying down the scaffolding for tissue repair and angiogenesis, by secreting fibronectin, laminin, and hyaluronate [3]. The MMP-mediated degradation of a peri-tumoural post-capillary venule facilitates the migration of endothelial cells and pericytes toward the pro-angiogenic stimulus created by the tumour cells [14,15]. Thus, the sprouting of new blood vessels is directed by the concentration gradient of pro-angiogenic factors, reaching the tumour [9].

Angiogenesis is directed towards the hypoxic tissue mainly by integrins, which promote endothelial cell migration and survival and pro-angiogenic macrophage trafficking to tumours [16]. Many tumours over-secrete vascular endothelial growth factor (VEGF), a highly potent pro-angiogenic factor, which in combination with the permeable micro-environment created by CAFs, serve as the perfect background for extensive vessel formation [17]. This excess of pro-angiogenic factors left unchecked by the lack of anti-angiogenic ones leads to abnormal, leaky vessel formation, which is a hallmark of malignant angiogenesis [17]. This results in a suboptimal blood flow, leading to a maintained state of hypoxia and further VEGF production in a vicious cycle. Additionally, the abnormal vessel formation is an ideal entry point of tumour cells into the circulation by providing an increased density of immature, leaky blood vessels that have little basement membrane and fewer intercellular junctional complexes than normal mature vessels [18].

## 2. Molecular Targets of Tumour Angiogenesis

As angiogenesis is a multi-factorial process, there are a multitude of potential targets for molecular imaging. These targets can be subdivided into direct targets (specific receptors on or near the cell that are directly related to angiogenesis) and indirect targets (glucose metabolism, hypoxia—which are only indirectly correlated with angiogenesis) (Table 1, Figure 1).

The discovery of VEGF has revolutionised the understanding of vasculogenesis and angiogenesis during embryogenesis and physiological homeostasis. It is the key mediator of angiogenesis and binds two VEGF receptors (VEGF receptor-1 and VEGF receptor-2), which are expressed on vascular endothelial cells [8,42]. It is up regulated by oncogene expression, a variety of growth factors and hypoxia. It is important to note that tumour vasculature formed in answer to VEGF influence is structurally and functionally abnormal: blood vessels are not organised into venules, arterioles, and capillaries, they are leaky and haemorrhagic.

Another key target for angiogenesis is integrins. The adhesive interaction of vascular endothelial cells is crucial in angiogenesis, with several adhesion molecules, including members of the integrin, cadherin, selectin, and immunoglobulin families as key effectors [43]. Integrins represent a family of cell-adhesion molecules crucial to cell-to-extracellular matrix and cell-to-cell interactions. Integrins are heterodimeric transmembrane glycoproteins composed of two non-covalently associated α and β subunits, the specific combination of these units determines signalling properties as well as binding affinity for ligands (and therefore tracers). Especially important are the α_v_ integrins, which are overexpressed on the surface of endothelial cells during angiogenesis. Different types of integrins can also imply different metastatic progresses: Hoshino et al. report that a high content of exosomal integrin α_v_β_5_ was associated with liver metastasis, whereas exosomal integrins α_6_β_4_ and α_6_β_1_ correlates with lung metastasis [44]. α*_v_*β_3_ integrin is one of the most studied integrins as it represents a highly specific biomarker to distinguish new from mature capillaries, allowing vascular mapping of angiogenesis in tumours [45,46].

Outside of the cell, MMPs are a relatively new and interesting target. MMPs are a family of zinc-binding metalloproteinases that participate in the degradation of the extracellular matrix, including the tumour capsule, resulting in tumour metastasis or invasion of the surrounding tissues. Furthermore, MMPs promote tumour growth and spread through the capillary endothelium [47]. MMPs can also promote neovascularisation by assisting in the breakdown of the extracellular matrix surrounding epithelial cells, resulting in weakened cell–cell tight junctions and adhesive connections and eventually migration of these epithelial cells [48,49].

There are also indirect techniques to measure angiogenesis, namely Glucose Transporter (GLUT) expression and hypoxia imaging, or even the recently discovered prostate specific membrane antigen (PSMA). As hypoxia is one of the key driving elements of angiogenesis, it makes sense to target it as a pseudo-target for angiogenesis. The degree of hypoxia if a strong prognostic factor as it is inversely correlated to survival: oxygen-deprived cells are highly resistant to therapy including radio- and chemotherapy, and survival of such cells is the primary cause of disease relapse. It is currently not well understood to what degree hypoxia markers can be directly used to visualise angiogenesis, partly because of suboptimal tracers and partly because successful angiogenesis implies a reduction in hypoxia [50].

Several authors demonstrated a co-upregulation in the expression of glucose transporters (GLUTs) and VEGF during 12-, 18-, and 24 h of severe hypoxia in vivo (xenografts) and in vitro (cell cultures), suggesting a modulation of the glucose kinetics by angiogenesis-related genes [51,52]. As GLUTs can be targeted with a commonly used tracer ([^18^F]-FDG), it is certainly worthwhile to investigate GLUTs as an angiogenesis marker. PSMA, a target well known for visualising prostate cancer, also seems to play a role in angiogenesis in solid tumours [53].

## 3. Indirect Targeting of Angiogenesis

### 3.1. [^18^F]-FDG

[^18^F]-FDG is a well-known and widely used tracer in the field of oncology (and beyond). Given the correlation between [^18^F]-FDG’s target GLUT1 expression and angiogenesis, several attempts have been made to use [^18^F]-FDG as a tool to monitor angiogenesis in vivo. Mirus et al. showed that both [^18^F]-FDG PET/CT and contrast enhanced CT are capable to detect parameters closely connected to the degree of tumour vascularization, and could therefore be used beyond its original role of quantifying glucose metabolism [54]. Strauss et al. provided some evidence for this hypothesis as [^18^F]-FDG kinetics were shown to be modulated by angiogenesis-related genes: the transport rate for [^18^F]-FDG (k1) is higher in tumours with a higher expression of VEGF-A and angiopoietin-2 [55]. More recently, Groves et al. observed in a study of 20 patients with early breast carcinoma, a correlation between mean standardised uptake value and endoglin (CD105, a marker for proliferation/angiogenesis) [56].

However, in comparison to direct angiogenesis tracers [^18^F]-FDG seems to be inferior. Provost et al. demonstrated this in a preclinical mouse tumour model using bevacizumab and temozolomide, where tumour uptake of [^68^Ga]-RGD, an integrin-specific angiogenesis tracer, was concordant with tumour growth in controls and in treated groups but [^18^F]-FDG was not [57].

Clinically, Durante et al. showed in a limited study of 10 patients that [^68^Ga]-NODAGA-RGD has a different spatial distribution than [^18^F]-FDG bringing different tumour information [38]. Toriihara et al. and Vatsa et al. each later confirmed this concept in other small studies, comparing [^18^F]-FDG with [^18^F]-FPPRGD_2_ [58,59]. Guo et al. investigated but found no correlation between micro vessel density (a proliferation-related endothelial cell marker that reflects active angiogenesis) and [^18^F]-FDG uptake using immunohistochemical staining measurements of angiogenesis in NSCLC [60]. This different distribution raises suspicions about the applicability of [^18^F]-FDG as a true angiogenesis marker.

[^18^F]-FDG does have a clinical use however, as it remains a tracer with strong prognostic power—even when used in reference to angiogenic treatment. Several studies provided evidence that pre-treatment [^18^F]-FDG PET can serve as an imaging biomarker for predicting survival following anti-angiogenic therapy with bevacizumab [61,62,63].

### 3.2. Hypoxia

Angiogenesis is recognised as an outcome of hypoxia, where signalling of proliferative markers such as VEGF is upregulated [64]. The degree of angiogenesis and vascular remodelling of vessels is one of the different parameters used to distinguish low-grade from high-grade tumours. Cher et al. already demonstrated, in 2006, that there was a strong correlation between tumour grade, angiogenesis markers and a PET hypoxia marker ([^18^F]-FMISO). [^18^F]-FMISO is a nitroimidazole compound, which may be intracellularly oxidised in normoxic cells. On the other hand, in hypoxic cells, FMISO cannot be oxidised, and it permanently binds to cellular components. The correlation of angiogenesis and hypoxia (imaging) was further investigated by several independent groups [65,66]. Ueda et al. showed in a bevacizumab effectiveness study that non-responders exhibited a higher degree of angiogenesis with more severe hypoxia than responders during treatment [67]. Similarly, in a study by Bekaert et al. [^18^F]-FMISO PET uptake was closely linked to tumour grade, with high uptake in glioblastomas [68]. Expression of biomarkers of hypoxia and angiogenesis markers (VEGF and others) were significantly higher in the [^18^F]-FMISO uptake group. Most importantly, this last study showed that patients without [^18^F]-FMISO uptake had a longer survival time than uptake positive patients—indicating a true clinical application.

However, even though VEGF is a major factor implicated in angiogenesis, evidence for VEGF expression in tumours correlating with hypoxia is conflicting [69]. For example, VEGF was ubiquitously expressed throughout tumours in a preclinical study regardless of proximity to capillaries or areas of necrosis. In addition, regions of severe hypoxia did not correlate with areas of upregulated VEGF expression [70].

[^18^F]-FMISO is a well-known hypoxia marker that has been shown to be useful as a prognostic marker, in glioma but also in other tumours [71,72], with an uptake mechanism proven to be hypoxia-dependent and glutathione conjugation [73]. Due to suboptimal pharmacokinetics, ^18^F-FMISO is not ideal for imaging hypoxia however, and alternative tracers are being investigated. For example, [^64^Cu]-diacetyl-bis-(*N*^4^-methylthiosemicarbazone) ([^64^Cu]-ATSM) has been shown to be superior in terms of imaging performance, and may even play a role as an add-on for anti-angiogenic tumour treatment [40,74,75,76]. Similar to FMISO, ATSM also demonstrates a strong correlation between the presence of angiogenic markers and the appearance of hypoxic regions.

An alternative hypoxia tracer is [^18^F]-FAZA, which has improved pharmacokinetics and biodistribution [77,78]. Like FMISO, it belongs to the family of nitroimidazole based tracers, which are typically reduced in hypoxic conditions, leading to intracellular binding of oxygen radicals and general trapping. Moreover, FAZA shows a higher tumour to background ratio [79].

[^18^F]-Flortanidazole ([^18^F]-HX4) was developed and validated both preclinically and clinically as a third-generation nitroimidazole-based tracer, with an even faster clearance and improved distribution [80]. Although we could not find any direct comparison of [^18^F]-HX4 and angiogenesis markers, its improved biodistribution and tumour visualization as demonstrated in several clinical trials make this a strong candidate for future (attempts at) imaging of angiogenesis in hypoxia-rich environments [81,82,83].

### 3.3. Matrix Metalloproteinases (MMPs)

It is also possible to target markers that are presented on the intercellular matrix. The main effort is on MMPs, as they are key proteolytic enzymes in tumour invasiveness [84]. MMPs are considered mediators of the alterations observed in the tumour microenvironment during cancer progression since they promote amongst others cancer cell signalling, migration, invasion, autophagy and angiogenesis [85]. There are currently 23 known MMPs, with the most focus currently on to the gelatinase-types (MMP-2 and MMP-9).

(2R)-2-[4-(6-[^18^F]Fluorohex-1-ynyl)-benzenesulfonylamino]-3-methylbutyric acid ([^18^F]-SAV03) was developed to visualise MMP-2 [23]. In 2003, they already demonstrated favourable tumour uptake, either by using the tracer directly or as a pro-drug. In 2015, another group labelled another MMP-2 tracer with ^68^Ga, showing moderate tumour to background ratios [24]. However, the degree of correlation to angiogenesis remains to be proven in the field of imaging and more research is needed.

### 3.4. Prostate Specific Membrane Antigen (PSMA)

PSMA is a well-known target for prostate cancer, using PET isotopes for imaging (^68^Ga, ^18^F) and β-emitters (^90^Y, ^177^Lu) for therapeutic applications of (metastasised) prostate cancer. Although PSMA is not expressed in non-prostate tumours, nor in healthy vasculature, expression does occur in endothelial cells of tumour-associated neovasculature [86]. Two excellent reviews of PSMA-based imaging in non-prostate imaging have been recently published [53,54,55,56,57,58,59,60,61,62,63,64,65,66,67,68,69,70,71,72,73,74,75,76,77,78,79,80,81,82,83,84,85,86,87], but we would like to briefly present the findings regarding angiogenesis here. The functional role of PSMA in angiogenesis is not clear at this moment, nor is PSMA a requirement for tumour-associated neovascularisation—but PSMA expression is associated to tumours that critically depend on angiogenesis. It has been proposed that PSMA exerts its exopeptidase activity on small peptides, originating from actions by to the endopeptidase matrix metalloproteinase-2, on its own already a candidate target in angiogenesis imaging [88]. The prognostic power of PSMA has been shown in squamous cell carcinoma of the head and neck, osteosarcoma, colorectal cancer, adenocarcinoma of the pancreas, lung cancer and others, demonstrating that PSMA expression appears to reflect prognostically relevant tumour features for several PSMA-expressing tumour entities [53]. However, this does not apply to all tumours; e.g., adrenocortical and gastric carcinomas did not show a correlation between PSMA uptake and tumour staging [53]. Moreover, PSMA has also been proposed for a dual diagnostic and therapeutic approach in triple negative breast cancer, for its ability to target multiple targets within the tumour microenvironment, including newly formed vessels [89].

In addition to the clinical imaging data providing information about the prognostic capability of PSMA, preclinical work has also been done indicating its critical role in regulating angiogenesis and progression of glioblastoma [90]. In this recent work, the authors also showed a possible therapeutic application (outside of prostate cancer).

## 4. Direct Targeting of Angiogenesis

### 4.1. VEGF

VEGF interacting with its receptor tyrosine kinase (VEGFR) is an important mediator of the angiogenesis pathway and consequently a potential imaging target [91,92,93]. The binding of VEGF to its receptor initiates a signalling cascade that promotes proliferation, migration, and survival of endothelial cells, leading to angiogenesis [94,95]. There are three endothelium-specific tyrosine kinases receptors, VEGFR-1, VEGFR-2 and VEGFR-3, of which mainly VEGFR-2 is overexpressed in a variety of solid tumour cells [8,96,97,98]. It has been shown that VEGF receptors (VEGFRs) are over-expressed in both grade IV glioma vasculature and grade IV glioma cells [99]. Its expression (as determined by histopathology) is shown to correlate negatively with overall patient survival.

VEGF consists of at least six isoforms of a various number of amino acids (121, 145, 165, 183, 189 and 206) produced through alternative splicing, and are active as homodimers linked by disulphide bonds [100]. VEGF121, VEGF165 and VEGF189 are the major forms secreted, of which mainly VEGF121 and VEGF165 are studied as potential tracers. The most studied tracer targeting VEGFR is ^123^I or ^125^I-labelled VEGF165/121, especially in the first decade of this millennium. However, also [^99m^Tc]-VEGF121, [^111^In]-VEGF165, [^64^Cu]-DOTA-VEGF121 and [^64^Cu]-VEGF121 have been investigated, as well as two radiolabelled versions of Bevacizumab: [^111^In]-Bevacizumab and [^89^Zr]-Bevacizumab.

The basic concept has been shown to work in vitro, as Li et al. already demonstrated in 2001 that [^123^I]-VEGF165 binds to a variety of human tumour cells/tissues compared with the corresponding normal tissues or normal peripheral blood cells [101]. They also demonstrated this for [^123^I]-VEGF121, although to a lesser extent. In the following years, several authors continued this work in preclinical models using non-invasive imaging with iodinated VEGF [27,102,103]. Backer et al. prepared a Cys-tagged vector of VEGF121 by cloning two single-chain of VEGF121 joining head-to-tail to express as scVEGF, which can be labelled as [^64^Cu]-1,4,7,10-tetraazacyclododecane-N,N′,N″,N‴-tetraacetic acid (DOTA)-scVEGF ([^64^Cu]-DOTA-scVEGF) or [^99m^Tc]-hydrazinonicotinic acid (HYNIC)-scVEGF ([^99m^Tc]-HYNIC-scVEGF) [104]. They later showed a good and specific tumour to background ratio in a preclinical model for both the ^99m^Tc and the ^64^Cu-labelled scVEGF, even though a high degree of heterogeneity was noticed [27]. Aiming for imaging at a later time point, Chan et al. developed a recombinant protein composed of VEGF165 fused through a flexible polypeptide linker to the n-lobe of human transferrin (hnTf), thus allowing labelling without the introduction of a specific chelator [105]. Although showing a reasonable tumour to background ratio, this tracer was found to be insufficiently stable in vivo. Cai et al. and Wang et al. further explored another variant labelled with ^64^Cu, and managed to reduce renal toxicity while maintaining VEGFR specificity [26,106].

Clinical trials with labelled VEGF so far are limited. One example in particular involving staging and follow-up of 18 patients with solid gastrointestinal tumours, [^123^I]-VEGF165 scans were compared with CT and MRI, demonstrating the usefulness of the [^123^I]-VEGF165 scan to visualise the tumour angiogenesis, despite the superiority of CT and MRI for the visualization of the gastrointestinal tumours and metastasis [25]. Li et al. further investigated biodistribution, safety, and dosimetry of [^123^I]-VEGF165 in nine patients with pancreatic carcinomas. They demonstrated that [^123^I]-VEGF165 visualised the primary tumour and their metastasis, but also noticed severe deiodination thus hampering routine clinical applicability [107]. Ubl et al. further tried to visualise B-cell lymphoma of the Mucosa-Associated Lymphoid Tissue (MALT lymphoma) using [^123^I]-VEGF165 in three patients but failed to visualise the tumours. More recently, the World Health Organization (WHO) grade IV glioma lesions were shown to have significant [^123^I]-VEGF uptake 18 h after the injection, whereas other brain tumours of grade II or III showed negative results in a study with 18 patients [108]. The most important finding was that [^123^I]-VEGF scans yielded prognostic information as patients with [^123^I]-VEGF T/N ratio above a particular threshold showed significantly longer survival than patients below that threshold (2680 vs. 295 days).

As few of these VEGF tracers went beyond the preclinical stage, an alternative pathway was explored: radiolabelled Bevacizumab. After preliminary mouse trials, [^89^Zr]-bevacizumab PET was able to visualise tumours in renal carcinoma patients [28,109]. High baseline tumour SUV_max_ was associated with longer time to progression, and Bevacizumab/interferon-α treatment strongly decreased tumour uptake. Even though the tracer showed success in visualising glioma and was able to distinguish grade 4 from lower grade glioma, the authors did show a high interpatient and intra-patient heterogeneity and further clinical application has been modest. The reason for this is the high heterogeneity of VEGFR expression in tumours, leading to disappointing clinical trials both for imaging as well as treatment (for example with bevacizumab) [110,111]. Studies are ongoing however; and similar to the [^123^I]-VEGF scans [^89^Zr]-bevacizumab may prove useful as a prognostic tool, even in children [112].

### 4.2. Integrins

Physiologically, healthy endothelial cells do not or little express integrins: integrins generally characterise a pathological state and are expressed on tumour cells and its associated neo-vasculature [113]. Among integrins, αvβ3 is highly expressed in tumours such as osteosarcomas, neuroblastomas, glioblastomas, malignant melanomas, breast, lung and prostate carcinomas and is, therefore, the prime target for imaging. The RGD motif, and the 3-dimensional structure of the ligand, is essential in binding to integrins.

#### 4.2.1. Fibronectin

Besides using the RGD-motif in the form of a small peptide integrins can also be targeted using the (much larger) radiolabelled (modified) fibronectin. In vitro and in vivo studies showed good biodistribution and tumour to background ratios for ^99m^Tc, ^123^I, ^124^I and ^76^Br labelled human recombinant anti-ED-B fibronectin antibody fragments, but clinical studies were generally not attempted [30,32,33,114].

One clinical study used [^123^I]-labelled dimeric L19 [L19(scFv)(2)] in 20 patients with lung, colorectal, or brain cancer [31]. This tracer showed a preference for aggressive, actively growing cancers, leading the authors to speculate about possible therapeutic applications.

#### 4.2.2. RGD-Motif

The RGD motif is a more “popular” agent to study integrin visualisation when compared to antibody fragments, as the small peptide is easier to produce and handle. Both linear and cyclic RGD peptides can theoretically be used as targeting biomolecules to develop α_v_β_3_-targeted radiotracers, but although linear peptides showed superior binding affinity in in vitro studies; this was reversed in vivo as well as showing lack of specificity. Most importantly, linear peptides are naturally unstable in vivo due to the high susceptibility to proteases [115,116]. The most promising RGD peptides allowing identification and location of the α_v_β_3_ integrin are small-sized, cyclic, and with a non-natural phenylalanine into the cyclic RGD sequence as these parameters increase affinity and stability (Figure 2).

The first tracers that have been developed were monomers of the RGD sequence. The first RGD adducts were obtained by electrophilic radio-iodination for SPECT imaging, showing correct specificities for α_v_β_3_ both in vitro and in vivo, but also a strong hepatic and intestinal uptake resulting from their hepatobiliary elimination, thus limiting their applicability [117]. ^99m^Tc-labelled compounds were also developed, with different bifunctional chelators. However, they were characterised by a disturbingly high lipophilicity that led to a strong hepatic uptake and elimination [118]. This high liver uptake proved again problematic in a clinical setting [119].

Several variants of RGD have been produced by adding groups to improve the biodistribution and pharmacokinetics. The [^18^F]-Galacto-RGD, which is RGD with an added glycosylation, is perhaps the most studied integrin-targeting tracer in PET and can be considered as a reference in the study of the expression of α_v_β_3_ [120]. This tracer presented a specific binding both in animal and human studies, fast blood clearance, good pharmacokinetic properties and an uptake corresponding to the expression of α_v_β_3_. Over the past two decades, [^18^F]-Galacto-RGD has been used in a multitude of preclinical and clinical trials, investigating, amongst others, malignant melanomas, glioblastomas, head and neck cancers, sarcomas, renal cell cancers, non-small cells lung cancers (i.e., NSCLC) and prostate cancers. Its main field of application was found to be in glioma, as [^18^F]-Galacto-RGD did not show a high uptake in normal brain tissue, especially when compared to [^18^F]-FDG [117]. Other variants followed: [^18^F]-Fluciclatide developed by GE Healthcare and [^18^F]-RGD-K5 developed by Siemens Molecular Imaging Inc. Although they all showed suitable properties in preclinical studies, their long and sometimes complicated synthesis made them unattractive as commercially viable tracers [117].

Next to glycosylation, pegylation was also attempted. When comparing [^18^F]-FB-RGD, a fluorobenzoyl-labelled RGD peptide with its pegylated analogue [^18^F]-PEG-RGD, Chen et al. found that pegylation resulted in an improved tumour uptake, but also a slowed pharmacokinetics and clearance [37]. [^18^F]-FPRGD2, an RGD-variant with a mini-pegylation, was found to show improved radiosynthesis yields, equal tumour uptake and reduced renal uptake in a preclinical study [121]. Later clinical studies confirmed its applicability [122,123,124]. Its uptake did not seem to correlate with that of [^18^F]-FDG.

Dimerization is another key modification to the RGD motif. Although not adapting the motif as such the inherent increased local concentration of RGD, when in presence of high concentrations of integrins, should lead to an improved affinity. [^18^F]-FPPRGD2, a dimeric peptide designed by conjugating [^18^F]-NFP with pegylated RGD dimeric peptide was also shown to bind to integrin-expressing tumours both in vitro and in vivo. A biodistribution study of [^18^F]-FPPRGD2 in human revealed promising pharmacokinetic properties [125]. When comparing [^18^F]-FPPRGD2 to [^18^F]-FDG, no difference was noted for primary lesions as well as metastases [126]. [^18^F]-FPPRGD2 was slightly superior, as it detected several small metastases that [^18^F]-FDG did not, while it proved negative in inflammatory lymph nodes where [^18^F]-FDG showed a false positive. Similarly, [^18^F]-FPPRGD2 PET showed a higher detection rate of recurrent glioblastoma multiform than that of brain MRI (100.0% vs. 93.3%) [127].

The concept of dimerization was taken one step further by synthesising a tetrameric compound [^18^F]FB-mini-PEG-E{E[c(RGDyK)]2}2; albeit without significant advantages both in vitro and in vivo [128]. Dimerization was also tried for ^99m^Tc and ^68^Ga compounds—again modifying the kinetics but generally maintaining tumour to background ratio [129,130,131]. The positive effect of multimerization on tumour uptake was also demonstrated in ^64^Cu-labeled octameric RGD-peptide as well as [^111^In]-labelled monomeric, dimeric and tetrameric analogues [132,133]. Even more octamer peptides have been studied with other isotopes, with similar good-but-not-superior results [134,135,136]. Cyclic mono or multimeric peptides, labelled with ^68^Ga through a chelator, may also prove to have a future, although a head-to-head clinical comparison is still missing [57,137,138,139,140]. Using the chelator as a positive concept in the RGD structure, some researchers even head back to ^18^F and investigate ^18^F-labelled NOTA-RGD variants [141].

The added clinical value of imaging integrins is still being investigated. For example, [^18^F]-FPPRGD2 provides additional information when compared to [^18^F]-FDG and may even outperform routine brain MRI [127]. In an atypical study in canines, Clemmensen et al. showed that [^68^Ga]Ga-NODAGA-E[(cRGDyK)]2 (RGD) PET and hyperpolarised [1-13C]pyruvate-MRSI (hyperPET) provide supplemental information and made a case for a more widespread use of combining angiogenesis imaging with energy metabolism to optimise patient treatment. A human clinical trial confirmed this case, as [^18^F]-RGD uptake on PET/CT imaging pre-treatment may predict the response to antiangiogenic therapy, with higher [^18^F]-RGD uptake in tumours predicting a better response to apatinib therapy [142].

The question remains which RGD peptide is the optimal choice. Both ^18^F, ^68^Ga and ^99m^Tc labelled tracers have shown good biodistribution and pharmacokinetics as well as tumour uptake in a variety of tumours [58,59,129,143,144,145]. The added value compared to [^18^F]-FDG is now emerging, especially in the field of head and neck tumours [146,147], but which tracer that should be integrated into daily clinical routine remains an open question.

### 4.3. NGR

Aminopeptidase N (APN/CD13), together with VEGF and RGD, is a key tumour angiogenesis marker. CD13 plays an important role in peptide cleavage, such as angiotensins, kinins, enkephalins, cytokines and chemokines. It also participates in extracellular matrix protein degradation, facilitating tumour cell invasion and migration.

Several studies reported that CD13 is overexpressed in the endothelial cells of tumour vasculature and in several solid tumours. Due to this increased expression, CD13 was reviewed as an important clinical marker in several inflammatory diseases and malignant cancers [148,149,150].

Peptides with an NGR (Asn-Gly-Arg) motif have been shown to have a high affinity for CD13 and several analogues have been synthesised over the past decade. The research field, starting with ^99m^Tc, quickly moved from monomeric to multimeric peptides as they showed improved affinity [39,151,152]. Several ^68^Ga and a ^64^Cu have been developed recently, each demonstrating good tumour to background ratios in preclinical models [41], [153,154,155]. Kis et al. compared radiolabelled NGR and RGD imaging in a preclinical tumour model and showed that [^68^Ga]-NOTAc(NGR) of the primary tumours was significantly higher than that of the accumulation of the commercially available [^68^Ga]-NODAGA-[c(RGD)]_2_ [40,153].

No NGR peptide had been tested in a clinical setting yet, so the clinical value has yet to be proven. In an interesting twist, Gai et al. recently made a dual receptor combining Integrin α_v_β_3_ and Aminopeptidase N Dual-Receptor. [^68^Ga]-NGR-RGD showed higher binding avidities, targeting efficiency and longer tumour retention time compared with monomeric [^68^Ga]-NGR and [^68^Ga]-RGD [156]. It may turn out that multimeric, multi-receptor peptides are the way to go for clinical applications.

## 5. Beyond Imaging

Driven by the successful clinical applications of [^177^Lu]-DOTA-TATE and [^177^Lu]-PSMA, both therapeutic agents developed based upon their ^68^Ga-predecessor designed for imaging, research is now also focusing to investigate the possible role of angiogenesis markers as potential therapeutic targets.

Ma et al. combined NGR with VEGF to produce a novel fusion protein labelled with the therapeutic isotope ^188^Re. ^188^Re has similar chemical characteristics as ^99m^Tc but emits an electron in addition to gamma lines that are suitable for SPECT—making the isotope suitable for radionuclide therapy. They demonstrated a favourable tumour to background ratio in HT-1080 tumour xenografts using SPECT, and more importantly showed that when applied in therapeutic concentrations [^188^Re]Re-NGR-VEGI showed excellent tumour inhibition effect with no observable toxicity [157]. Zhao et al. combined RGD with Evans blue and subsequently labelled this peptide analogue with ^177^Lu, another therapeutic isotope. [^177^Lu]-EB-RGD was shown to have an inhibitory effect on tumour growth, either on its own or in combination with other drugs [158,159]. Translation to clinical routine is not yet a possibility, but the concept of angiogenesis-markers labelled with therapeutic isotopes does have a future.

## 6. Materials and Methods

The information for this review was compiled from PubMed listed publications, using the following search terms (final search in March 2021):Selected reviews: PET imaging angiogenesis (353 results) and SPECT imaging angiogenesis (169 results).Original research: PET NGR (15 results), SPECT NGR (7 results), PET VEGF (413 results), SPECT VEGF (153 results), PET RGD (438 results), SPECT RGD (138 results), SPECT EGF (81 results), PET EGF (79 results), PET Fibroblast growth factor (132 results), SPECT Fibroblast growth factor (42 results), PET PDGF (39 results), SPECT PDGF (6 results), SPECT Angiopoietin-1 (3 results), PET Angiopoietin-1 (5 results), SPECT ephrin (2 results), PET ephrin (12 results), PET MMP angiogenesis (14 results) and SPECT MMP angiogenesis (6 results).

The resulting publication lists were subsequently selected only for oncology related publications in animals/humans, disregarding any findings regarding cardiology or vascular plaques and disregarding publications focusing purely on new radiochemical or chemical synthesis pathways.

## 7. Conclusions

Significant progress has been made over the past decade concerning molecular imaging in angiogenesis. While indirect imaging of angiogenesis using either [^18^F]-FDG or hypoxia-markers is not always strictly correlated to angiogenesis, they do have strong prognostic power in the clinical setting. Direct imaging with specific markers, targeting specific receptors on the cancer cells or the surrounding vasculature, are more tightly correlated to angiogenesis. Their clinical impact remains to be determined however, as there are a multitude of potentially useful tracers, both preclinical and clinical, but further research is needed to determine which of those will be translated to the clinical setting. In our opinion, the RGD family holds the most potential to enter clinical routine as a tracer for direct quantification of angiogenesis, by providing an improved assessment of tumour characteristics required for personalised therapy approaches.

## Figures and Tables

**Figure 1 ijms-22-05544-f001:**
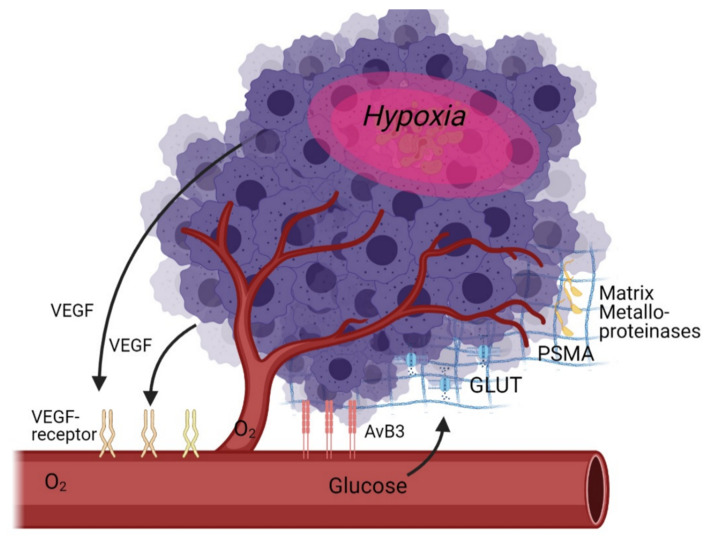
Schematic overview of angiogenesis targets in the tumour microenvironment as discussed in this review.

**Figure 2 ijms-22-05544-f002:**
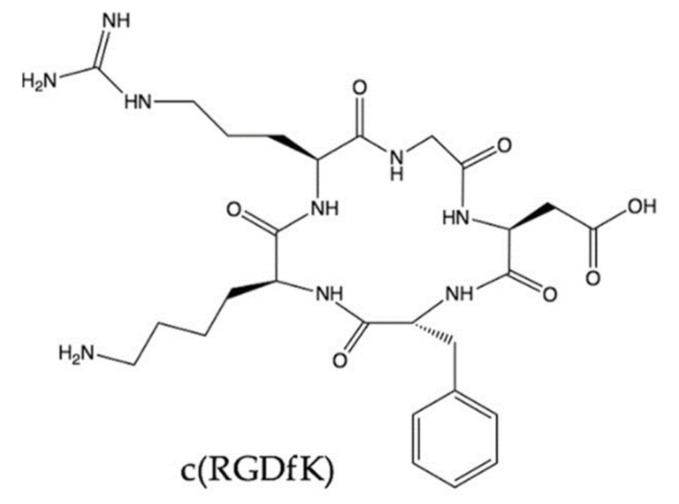
Cyclo(-RGDfK), a potent and selective inhibitor of the α_v_β_3_ integrin, with an IC_50_ of 0.94 nM.

**Table 1 ijms-22-05544-t001:** Classification of the most promising tracers for angiogenesis imaging.

Imaging Method	Target	Tracer Name	Modality	Stage
Indirect targeting of angiogenesis	Glucose metabolism	[^18^F]-FDG	PET	accepted for clinical use
	Hypoxia	[^18^F]-HX4	PET	clinical trial [19]
		[^18^F]-FMISO	PET	clinical trial [20]
		[^18^F]-FAZA	PET	clinical trial [21], [22]
	MMPs	[^18^F]-SAV03	PET	in vivo preclinical stage [23]
		[^68^Ga]-NOTA-C6	PET	in vivo preclinical stage [24]
Direct targeting of angiogenesis	VEGF	[^123^I]-VEGF165	SPECT	clinical trial [25]
		[^64^Cu]-DOTA-scVEGF	PET	in vivo preclinical stage [26]
		[^99m^Tc]-HYNIC-scVEGF	SPECT	in vivo preclinical stage [27]
		[^89^Zr]-Bevacizumab	PET	in vivo preclinical stage [28]
		[^111^In]-Bevacizumab	SPECT	clinical trial [29]
	Integrins	^99m^Tc-labelled anti-ED-B fibronectin antibody	SPECT	in vivo preclinical stage [30]
		^123^I-labelled anti-ED-B fibronectin antibody	SPECT	in vivo preclinical stage [31]
		^124^I-labelled anti-ED-B fibronectin antibody	SPECT	in vivo preclinical stage [32]
		^76^Br-labelled anti-ED-B fibronectin antibody	PET	in vivo preclinical stage [33]
		[^123^I]-L19(scFv)	SPECT	clinical trial [31]
		[^18^F]-Galacto-RGD	PET	clinical trial [34]
		[^18^F]-Fluciclatide	PET	clinical trial [35]
		[^18^F]-RGD-K5	PET	clinical trial [36]
		[^18^F]-FB-RGD	PET	in vivo preclinical stage [37]
		[^18^F]-PEG-RGD2	PET	in vivo preclinical stage [37]
		[^68^Ga] Ga-NODAGA-RGD	PET	clinical trial [38]
	NGR	^99m^Tc-labelled NGR	SPECT	in vivo preclinical stage [39]
		[^68^Ga]-NOTA-c(NGR)	PET	in vivo preclinical stage [40]
		^64^Cu-labelled NGR	PET	in vivo preclinical stage [41]

## Data Availability

Data sharing is not applicable. No new data were created or analysed in this study.

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
