# Peer review of "Molecular Imaging of Angiogenesis in Oncology: Current Preclinical and Clinical Status"

_ijms, 2021, doi:10.3390/ijms22115544_

Round 1

Reviewer 1 Report

This review gives an overview of the radio tracers used for imaging angiogenesis in oncology (preclinical and clinical papers). The review well organized and comprehensively described. I don't have any major issues. A nice table is also gives of the most promising radio tracers for angiogenesis imaging. 

Some minor issues are:

1) In the table only 18F-FMISO is mentioned as hypoxia tracer. Why are not FAZA or the third-generation hypoxia tracer HX4 mentioned?

2) On page 5, line 160: I think it should be endoglin (= CD105) and not endoglodin

3) a general remark: in vitro/in vivo in Italic (is sometimes journal dependent)?

4) Page 6, paragraph 3.2 (hypoxia): no studies are mentioned of 18F-HX4? 

5) Page 7, line 240: ..PSMA expression  appears to reflect prognostically relevant tumor features for several, not all PSMA-expressing entities. What do the authors mean with this? Which relevant tumour features? Is a bit vague now. 

Reviewer 2 Report

In the review article entitled “Molecular imaging of angiogenesis in oncology: current pre-clinical and clinical status” Florea et al, put together different imaging techniques to monitor angiogenesis especially during tumor growth. This article can be very helpful to perceive knowledge about the “best use” of certain imaging techniques. This can further help to understand the complexity of blood vessel formation during tumorigenesis and can assist in postulating better therapeutic strategies for cancer.

While this is a nicely presented article, I have some concerns,  that should be taken care of to improve the article. Please see the following:

  1. In the abstract authors should identify why specifically this review is important in the field, Also, what are the conclusion and future in the field. “There is currently no 26 direct tracer that can be singled out as “most promising” this sentence can not be a concluding line for the abstract.
  2. In the introduction, the background should be elaborated
  3. References should be added in Table 1.
  4. Page 3 number 2 the subheading should be changed, as all the techniques discussed are already known and mostly in use,
  5. Elaborate on conclusion and future direction.

Round 2

Reviewer 2 Report

Thanks for incorporating the suggestions.